# *KCNQ1* p.D446E Variant as a Risk Allele for Arrhythmogenic Phenotypes: Electrophysiological Characterization Reveals a Complex Phenotype Affecting the Slow Delayed Rectifier Potassium Current (IKs) Voltage Dependence by Causing a Hyperpolarizing Shift and a Lack of Response to Protein Kinase A Activation

**DOI:** 10.3390/ijms25020953

**Published:** 2024-01-12

**Authors:** Antonia González-Garrido, Omar López-Ramírez, Abel Cerda-Mireles, Thania Navarrete-Miranda, Aranza Iztanami Flores-Arenas, Arturo Rojo-Domínguez, Leticia Arregui, Pedro Iturralde, Erika Antúnez-Argüelles, Mayra Domínguez-Pérez, Leonor Jacobo-Albavera, Alessandra Carnevale, Teresa Villarreal-Molina

**Affiliations:** 1Laboratorio de Enfermedades Mendelianas, Instituto Nacional de Medicina Genómica (INMEGEN), Mexico City 14610, Mexico; antonia.gonzalez-garrido@cuanschutz.edu (A.G.-G.); mireles.abel26@gmail.com (A.C.-M.); thania.navarrete.04bp@gmail.com (T.N.-M.); 315179789@cuautitlan.unam.mx (A.I.F.-A.); acarnevale@inmegen.gob.mx (A.C.); 2Instituto de Oftalmología Fundación de Asistencia Privada Conde de la Valenciana, I.A.P., Mexico City 06800, Mexico; omar.lpz.r@gmail.com; 3Departamento de Ciencias Naturales, Universidad Autónoma Metropolitana, Unidad Cuajimalpa, Mexico City 05348, Mexico; arojo@correo.cua.uam.mx (A.R.-D.); arregui.leticia@gmail.com (L.A.); 4Departamento de Electrofisiología, Instituto Nacional de Cardiología “Ignacio Chávez”, Mexico City 14080, Mexico; pedroi@yahoo.com; 5Instituto de Genética, Universidad del Mar, Campus Puerto Escondido, Oaxaca 70985, Mexico; eantunezarguelles@gmail.com; 6Laboratorio de Genómica de Enfermedades Cardiovasculares, Instituto Nacional de Medicina Genómica (INMEGEN), Mexico City 14610, Mexico; mdominguez@inmegen.gob.mx (M.D.-P.); ljacobo@inmegen.gob.mx (L.J.-A.)

**Keywords:** *KCNQ1*, p.D446E, low-frequency polymorphism, long QT syndrome, early repolarization, loss of function, gain of function, IKs

## Abstract

Genetic testing is crucial in inherited arrhythmogenic channelopathies; however, the clinical interpretation of genetic variants remains challenging. Incomplete penetrance, oligogenic, polygenic or multifactorial forms of channelopathies further complicate variant interpretation. We identified the *KCNQ1*/p.D446E variant in 2/63 patients with long QT syndrome, 30-fold more frequent than in public databases. We thus characterized the biophysical phenotypes of wildtype and mutant IKs co-expressing these alleles with the β-subunit minK in HEK293 cells. *KCNQ1* p.446E homozygosity significantly shifted IKs voltage dependence to hyperpolarizing potentials in basal conditions (gain of function) but failed to shift voltage dependence to hyperpolarizing potentials (loss of function) in the presence of 8Br-cAMP, a protein kinase A activator. Basal IKs activation kinetics did not differ among genotypes, but in response to 8Br-cAMP, IKs 446 E/E (homozygous) activation kinetics were slower at the most positive potentials. Protein modeling predicted a slower transition of the 446E Kv7.1 tetrameric channel to the stabilized open state. In conclusion, biophysical and modelling evidence shows that the *KCNQ1* p.D446E variant has complex functional consequences including both gain and loss of function, suggesting a contribution to the pathogenesis of arrhythmogenic phenotypes as a functional risk allele.

## 1. Introduction

Long QT syndrome (LQTS) is the most frequent arrhythmogenic channelopathy. Because the patients may develop malignant arrhythmia and sudden cardiac death, the discovery of causal mutations is crucial for the identification of family members at risk for sudden cardiac death and may have implications in clinical management. Approximately 75% of the cases are caused by mutations in three major genes: *KCNQ1*, *KCNH2* and *SCN5A* [1]. *KCNQ1* encodes the Kv7.1 channel, an oligomeric protein consisting of an α pore-forming subunit (Kv7.1) encoded by *KCNQ1*, and the β accessory subunit minK encoded by *KCNE1.* Kv7.1 and minK are responsible for the slow delayed rectifier potassium current (IKs), one of the currents that contributes to the repolarization phase of cardiac action potential, particularly under increased β-adrenergic stimulation through the activation of the protein kinase A (PKA) [2,3]. While most *KCNQ1*-associated LQTS cases are inherited in an autosomal dominant fashion, homozygous or compound heterozygous *KCNQ1* mutations cause the autosomal recessive Jervell and Lange-Nielson syndrome (with a marked QT prolongation and congenital sensorineural deafness) or autosomal recessive Romano–Ward syndrome, without deafness but with severe QT interval prolongation [1,4].

Over the last decade, genetic testing has become part of the diagnostic work-up for many inherited diseases. However, the burden of variants classified as of uncertain significance (VUS) by the 2015 American College of Medical Genetics guidelines (ACMG) [5] is still a significant challenge for clinicians and geneticists. In the case of arrhythmogenic channelopathies, variants of incomplete penetrance, acquired channelopathies and different forms of inheritance (dominant, recessive, oligogenic and polygenic) further complicate the clinical interpretation of rare and even common genetic variants [6,7]. In addition, low, or very low frequency missense variants may cause infrequent recessive forms of the diseases or partially alter the protein function, contributing to the etiology of disease in the presence of other functional variants or environmental challenges [8,9]. Giudicessi et al. proposed the concept of “functional risk alleles” as variants with strong epidemiological and functional evidence to suggest a pathogenic contribution but with a minor allele frequency greater or equal to disease prevalence or ≥5% in public exomes which are criteria of benign variants [10]. Thus, these variants are not considered as causal of Mendelian forms of the disease but may contribute to the etiology of the arrhythmogenic syndromes. 

Next-generation sequencing (NGS) of Mexican patients with LQTS has revealed the presence of several low or very low frequency variants in LQTS-associated genes that are most frequent or found exclusively in Latin American populations. Such is the case of *KCNQ1* p.A300T (rs120074187), which is almost 10 times more frequent in Latinos (MAF 2.61 × 10^−4^) than in all gnomAD alleles (MAF 4.8 × 10^−5^) (https://gnomad.broadinstitute.org/variant/11-2594193-G-A?dataset=gnomad_r2_1, accessed on 7 November 2023). While this variant is reported as having conflicting evidence of pathogenicity in ClinVar (https://www.ncbi.nlm.nih.gov/clinvar/RCV000003276/, accessed on 7 November 2023), clinical evidence and electrophysiological characterization have shown that it is in fact functional and is a recessive allele causing Romano–Ward syndrome when found in homozygosity or compound heterozygosity [4,11,12]. Similarly, the *KCNQ1* p.D446E variant (rs199472780) is a very low frequency variant classified as a VUS, which to date has been found in the gnomAD database only in Latinos (https://gnomad.broadinstitute.org/variant/11-2610029-C-G?dataset=gnomad_r2_1, accessed on 7 November 2023). We found this variant in 2/63 unrelated Mexican patients with diagnosis or suspected LQTS. The allele frequency of *KCNQ1* p.D446E was 30-fold higher in our study group (MAF = 0.016) than that reported in gnomAD Latinos (MAF = 5.21 × 10^−4^). We thus analyzed the biophysical phenotype of this variant in the heterozygous and homozygous states, under basal conditions and after protein kinase A (PKA) activation simulating adrenergic stimulation and used protein modelling to understand the possible contribution of this variant to the etiology of arrhythmogenic phenotypes, as suggested by Giudicessi et al. as a functional risk allele [10].

## 2. Results

### 2.1. Patients

No pathogenic or likely pathogenic variant was found in 13/63 LQTS patients (20.6%). Two of these thirteen patients were heterozygous for the *KCNQ1* p.D446E variant (rs199472780). The minor allele frequency was 30-fold higher in our study group (0.016) than in Latinos from gnomAD (5.21 × 10^−4^, *p* = 0.00001). 

Index case 1 was a previously asymptomatic 16-year-old male who suffered generalized seizures followed by cardiac arrest while sitting in a classroom, which reverted after resuscitation maneuvers. He was then referred to the Instituto Nacional de Cardiología “Ignacio Chávez”. Basal ECG showed a normal QTc interval (460 ms) and early repolarization (ER) in leads V2 to V5 (Figure 1). However, an electrophysiological study showed a clearly prolonged QT interval (QTc 520 ms) and a U wave after adrenalin injection, without inducing ventricular tachycardia or fibrillation. Family history of sudden death was positive (Figure 2), and the Schwartz score indicated a high probability of LQTS. NGS revealed the index case was heterozygous for three additional low-frequency variants in channelopathy-associated genes that are more frequent in Latinos: *SCN5A* p.V1951L (rs41315493), *ANK2* p.G2221E (rs185384934) and *TRDN* p.S80F (rs181287533). All these variants were classified as benign or likely benign according to ACMG criteria; only *KCNQ1* p.D446E is considered a VUS. Index case 2 was an 8-year-old girl who was referred for cardiological evaluation because her mother died suddenly in the postpartum period (Figure 2). Although the girl was asymptomatic and had a normal QTc interval on the ECG at rest (QTc 380 ms), a Viskin test induced a mildly prolonged QT interval (QTc 478 ms). The Schwartz score indicated an intermediate probability of LQTS. Multiplex ligation-dependent probe amplification (MLPA) analyses ruled out large insertions or deletions in *KCNQ1*, *KCNH2*, *KCNE1* and *SCN5A* in both patients. 

### 2.2. The Homozygous KCNQ1-446 E/E Channel Left-Shifts IKs Voltage Dependence and Impairs Its Response to PKA Activation

Figure 3A shows examples of traces for each *KCNQ1* genotype without and with 8Br-cAMP. Figure 3B shows the activation curves of IKs-WT (446 D/D), IKs-446 D/E and IKs-446 E/E. The voltage dependence (V_1/2_) of IKs-446 E/E was significantly left-shifted (12.74 ± 1.83 mV) compared with IKs-WT (27.47 ± 1.57 mV, *p* = 1.75 × 10^−4^) and IKs-446 D/E (26.14 ± 3.55 mV, *p* = 0.006; Figure 3C). IKs-446 E/E also displayed a higher voltage sensitivity (15.30 ± 0.82 mV^−1^) than IKs-WT (10.96 ± 0.85 mV^−1^) and IKs-446 D/E (14.64 ± 1.12 mV^−1^), although only the difference with IKs-WT reached statistical significance (*p* = 0.005). No differences in terms of conductance density (G/Cm) were observed among genotypes (WT or 446 D/D, 446 D/E and 446 E/E) (*p* > 0.56). 

8Br-cAMP induced a significant left shift in voltage dependence in IKs-WT and IKs-446 D/E (−8.70 ± 3.81 mV, *p* = 3.56 × 10^−8^; and −7.55 ± 3.03 mV, *p* = 2.12 × 10^−8^, respectively), but not in IKs-446 E/E (5.29 ± 3.36 mV, *p* = 0.50) (Figure 3B,C). Similarly, 8Br-cAMP induced higher voltage sensitivity in IKs-WT (8.08 ± 0.83 mV^−1^, *p* = 2.64 × 10^−5^) and IKs-446 D/E (9.22 ± 0.76 mV^−1^, *p* = 0.007), but not in IKs-446 E/E (9.87 ± 0.88 mV^−1^, *p* = 0.98; Figure 3C). No mean differences in conductance density in response to 8Br-cAMP were observed among genotypes (*p* > 0.43).

### 2.3. KCNQ1-446 E/E Slows Activation Time of IKs in Response to 8Br-cAMP and Extends Its Activation Voltage Range to Negative Values 

Figure 4 compares IKs rise time values from all experimental series. In basal conditions (without 8Br-cAMP), IKs-446 E/E showed a broader activation voltage range as compared to IKs-WT, while IKs-446 D/E had an intermediate activation range, between that of 446 D/D and 446 E/E. IKs-446 D/E and IKs-446 E/E currents became activated at more negative potentials than IKs-WT, although the rise time values among genotypes did not show statistically significant differences (*p* > 0.6, Appendix A). Notably, only IKs-446 E/E showed rise time values at negative potentials and broadened the activation voltage range (Figure 4A). Under 8Br-cAMP stimulation, rise times values were faster at more depolarizing potentials in all genotypes. However, IKs-446 E/E showed slower rise times than IKs-WT at 24.5, 34.5, 44.5 and 54.5 mV (*p* = 0.002, 6.56 × 10^−4^, 0.015 and 0.02, respectively) and slower rise times than IKs-446 D/E at 24.5 and 34.5 mV (*p* = 0.003 and 0.01, respectively). No differences in rise time between IKs-WT and IKs-446 D/E were observed at any voltage (*p* > 0.067; Appendix A).

### 2.4. The KCNQ1 D446E Variant Does Not Affect IKs Deactivation Rate

Representative traces of all experimental series are shown in Figure 5A. Deactivations rates tested at −125.5 mV in basal conditions were not significantly different among genotypes (*p* > 0.88; Figure 5B). 8Br-cAMP significantly increased the mean deactivation time constant in all genotypes (*p* < 0.004), although no significant differences in the response to 8Br-cAMP were observed among genotypes (Figure 5B).

### 2.5. Kv7.1-D446E Modeling

Figure 6 depicts the structural configurations of the tetrameric 466E Kv7.1 channel in both the attached (green, calmodulin is in contact with the voltage sensor domain or VSD) and detached (red, calmodulin loses contact with the VSD) conformations, as previously outlined by Ma et al. [13], including our modelled domain. According to Ma et al.’s findings, both conformations feature open-channel structures, yet they differ in the relative positioning of the calmodulin-binding domains with respect to the VSD. To enhance clarity, Figure 6A exclusively showcases isolated monomers superimposed in both conformations, highlighting calmodulin binding in orange for the attached conformation and in yellow for the detached conformation. Appendix A shows monomers and tetramers without calmodulin for clarification. Notably, domain 5 undergoes not only a positional change but also a substantial 180° swiveling motion driven by the calmodulin-binding domain. This rotation facilitates the movement of residue 446 through the channel’s vestibule. The 466E mutant exhibits a more solvent-exposed electric charge compared to the wildtype aspartate due to an additional degree of side-chain freedom, resulting in approximately three times the volume of potential conformations (Figure 6C). The increased volume of the 446E residue is predicted to cause slower tetramer rotation movements as compared to the wildtype during the attached–detached equilibrium of open-channel conformations. Residue 446E is also predicted to interact with different components of the molecular system than 446D because of its increased volume. The presence of a longer glutamate side chain neighboring the channel vestibule would allow this residue to interact with a larger number of potassium cations, possibly reducing the effective local cation concentration. The amino and carboxyl domains each include more than 100 amino acids and have not been determined.

## 3. Discussion

Genetic testing in LQTS has a high diagnostic yield (75–80%). It has been suggested that cases where no pathogenic or likely pathogenic mutation is identified may be caused by deep intronic mutations, mutations in other yet unknown causal genes, or may be of digenic, oligogenic, polygenic or multifactorial etiology. In this regard, low or very low frequency genetic variants may play a role in non-Mendelian forms of the disease, representing a problem for molecular diagnosis, particularly when functional studies of the consequences of the variant have not been performed. In the case of ion channels causing arrhythmogenic phenotypes, the biophysical characterizations of these mutations usually only report voltage dependency in basal conditions in the homozygous state and less frequently report the characterization under PKA stimulation or comparisons between homozygous and heterozygous models. 

To date, in the gnomAD database, the *KCNQ1* p.D446E variant is found only in Latino populations with a very low frequency and is considered a variant of uncertain clinical significance. While the significantly higher frequency of this variant in our study group suggests it may play a role in LQTS as a risk allele, it cannot be considered a causal allele because of its frequency, and because not all carriers showed LQTS or ER. Moreover, although members from both families showed ER in anterior ECG leads (Appendix A), several noncarriers in family 1 also showed ER on anterior leads, suggesting the presence of other variants or environmental factors playing a role in the phenotypes. We thus sought to characterize the biophysical properties of this variant and to model the *KCNQ1* p.D446E tetramer to provide functional evidence suggesting it is a risk allele. 

### 3.1. KCNQ1-p.D446E Left-Shifts the IKs Activation Curve Suggesting a Possible Role in Early Repolarization

Firstly, *KCNQ1* 446E coexpressed with *KCNE1* did not suppress IKs function but shifted the voltage dependence of IKs-446 E/E to hyperpolarizing potentials, showed a higher voltage sensitivity than IKs-WT, and showed a broader activation voltage range. Hence, in the homozygous state this variant causes a gain of IKs function, which is compatible with a role in ER. It is estimated that around 5% of the population shows ER on the ECG [14]; however, this electrocardiographic feature has also been associated with ventricular fibrillation and postulated as a predictor to develop life-threatening ventricular arrhythmias [15,16,17]. Patients with ER show current imbalances between epi- and endocardial layers in a dispersion of de- and repolarization that might be the result of increased outward potassium currents, mainly the transient-outward K^+^ (I_to_) and adenosine triphosphate-sensitive current (IK_ATP_) or decreased inward depolarizing currents (sodium I_Na_, and calcium I_CaL_) [18]. Although *KCNQ1* gain-of-function mutations have been described in short QT syndrome [19], to our knowledge there is only one report of a heterozygous *KCNQ1* mutation also located within D5 (p.A399S) in a young man with ER and sudden cardiac death, without establishing the causal role of the mutation [20].

While the voltage sensitivity of both IKs-446 D/E and IKs-446 E/E was higher than that of IKs-WT, the difference between the wildtype and heterozygous forms did not reach statistical significance (*p* = 0.08). Thus, the *KCNQ1*-p.D446E left voltage shift cannot be strictly considered a dominant trait. This is relevant because no *KCNQ1* p.D446E homozygous individuals were found. However, the data suggest the heterozygous mutation may be a risk factor that contributes to arrhythmogenicity. 

### 3.2. KCNQ1-p.D446E Impairs the IKs Response to 8Br-cAMP (Loss of Function) Suggesting a Role in LQTS

In normal conditions, β-adrenergic stimulation facilitates the repolarization of the cardiac action potential via PKA activation [21]. Β-adrenergic stimulation shifts the voltage dependence to hyperpolarizing potentials, thus speeding IKs activation kinetics and enhancing channel activation during depolarization. It also increases the repolarizing current contributing to a rate-dependent decrease in action potential duration [22]. Interestingly, our experimental data showed that Iks-446 E/E voltage dependence did not change with 8Br-cAMP.

IKs activation kinetics were assessed by estimating the rise time. In basal conditions, IKs activation kinetics were not significantly different among genotypes (WT, 446 D/E and 446 E/E). As expected, 8Br-cAMP induced faster activation kinetics in all genotypes. However, while 8Br-cAMP induced a faster rise time in the entire IKs-WT activation range, 8Br-cAMP induced a complex behavior in the IKs-446 D/E activation rate, with a slower activation rise time at the most negative potentials, −5.5 and 4.5 mV (loss of function) but a faster rise time at most positive potentials, 34.5, 44.5 and 54.5 mV (gain of function). Moreover, 8Br-cAMP induced a faster IKs-446 E/E activation rise time only at 44.5 and 54.5 mV (loss of function at more negative potentials). Altogether, the absence of a shift in voltage dependence of IKS-446 E/E and the slower rise time values of IKs-446 D/E in response to 8Br-cAMP suggest the variant may contribute to the arrhythmogenic phenotypes. Notably, both our index cases showed normal QTc intervals in basal conditions, which became prolonged in response to adrenergic stimulation in index case 1, and in response to increased heart rate induced by a Viskin test in case 2. The latter is compatible with several reports of *KCNQ1* mutations with impaired β-adrenergic response associated with LQTS [23,24,25,26]. Thus, while in basal conditions the hyperpolarizing shift can be interpreted as gain of function, under 8Br-cAMP stimulation there is a loss of function. This may contribute to the phenotype complexity and variability observed in heterozygous carriers. 

### 3.3. Kv7.1-446E Modeling Predicts Slower Transition to the Stabilized Open State of the Channel

The *KCNQ1* p.D446E variant lies within D5 located in the carboxyl terminal domain of Kv7.1. D5 acts as a hinge between helices A and B, which in turn binds calmodulin and includes sites for post-translational modifications regulating the channel function [27,28]. Since the aspartate/glutamate substitution is conservative in electric nature but with broader mobility and increased solvent-exposed negative charge, its effects are predicted to be subtle in altering the gating kinetics. In this regard, high-resolution cryoelectron microscopy studies of the full-length Kv7.1–calmodulin complex have revealed intricated interactions between Kv7.1–VSD, calmodulin and the PIP2 complex [13,29]. The VSD transition to the fully activated state involves PIP2 competing with calmodulin for the same binding site. When PIP2 is absent, the Kv7.1–carboxyl terminal domain bends at the S6-HA linker to enable calmodulin interaction with the VSD at the S2–S3 linker (attached conformation). In the presence of PIP2, the Kv7.1–carboxyl terminal domain unbends to disengage calmodulin from the VSD (detached conformation), and the S2–S3 linker instead interacts with PIP2. The latter conformation triggers a large conformational change leading to the dilation of the Kv7.1 gate and stabilization of its open state [21,29]. The larger volume of the 446E residue predicts a slower transition of the Kv7.1 tetrameric channel from the attached to detached conformations, compatible with an impaired response to adrenergic stimulation. Moreover, although the presence of a longer glutamate side chain neighboring the channel vestibule predicted an increased interaction with potassium cations, possibly reducing the effective local cation concentration, IKs-446 E/E did not show any reduced conductance.

Summarizing, our model predicts that the interactions of the 466E negative charge with K^+^ ions and other groups in the surrounding environment may be stronger than those of the WT aspartate. This could affect the kinetics of the channel response to regulatory processes including the hinge movement of helices A and B for calmodulin recognition, VSD–PD (voltage sensor domain–pore domain) electromechanical coupling and possibly the phosphorylation of S27 and S92 sites responsible for the adrenergic modulation of IKs function [13,29,30,31]. Future investigations delving into these processes should contribute to a deeper understanding of the impact of this amino acid substitution on overall protein behavior.

Several study limitations should be pointed out. Firstly, while HEK293 cells have been widely used as a model to study voltage-gated ion channels, they do not reflect the physiological environment of cardiomyocytes. In addition, the study was conducted at room temperature (25 °C) and not at physiological temperature. Temperatures below 37 °C are known to decrease ion-channel current kinetics. Moreover, 8Br-cAMP was tested in the intracellular solution rather than extracellularly, with the purpose of obtaining a better control of the 8Br-cAMP concentration. In consequence, the effects were assessed in different cells rather than the same cell. Because of this, statistical differences were calculated by a two-way-ANOVA followed by a Tukey post hoc test of significance.

In conclusion, our results provide the first in vitro biophysical evidence of the impact of the *KCNQ1* p.D446E variant on cardiac IKs. These functional data and protein modelling suggest that the *KCNQ1* p.D446E variant could have complex physiological consequences, particularly in response to adrenergic stimulation. Thus, this variant, along with other functional alleles and/or environmental challenges, may contribute to the pathogenesis of ER and LQTS, as what has been called a functional risk allele [10]. More integrative experimental models such as iPSC-derived cardiomyocytes will lead to a better understanding of the contribution of this and other low-frequency variants, alone or in combination, to arrhythmogenic phenotypes, under adrenergic and other stimuli.

## 4. Materials and Methods

### 4.1. Subjects

We studied 63 patients with clinical diagnosis or suspected LQTS from the National Institute of Cardiology “Ignacio Chávez” in Mexico City, who were referred to the National Institute of Genomic Medicine for genetic testing. The Schwartz score was >3.5 (high probability) in 60 patients, and between 2 and 3 (intermediate probability) in 3 patients. 

### 4.2. Molecular Diagnosis

Genomic DNA was extracted from peripheral blood leukocytes with the DNeasy Blood Kit (Qiagen, Germantown, MD, USA). Sequencing was performed using the Trusight Cardio sequencing panel in a MiSeq System (Illumina, San Diego, CA, USA). Quality was assessed with FastQC version 11.1 (Babraham Bioinformatics, Babraham, UK), BWA Enrichment v2.1 was used for alignment, GATK v4.0 for variant calling and ANNOVAR (wannovar.wglab.org, accessed on 12 February 2021) was used for annotation. All novel or very low frequency variants (MAF < 0.0005) found in channelopathy-related genes were classified according to the criteria described by the ACMG [5]. 

### 4.3. Site-Directed Mutagenesis

*KCNQ1* (NM_000218) tagged with GFP and *KCNE1* (NM_001127670) tagged with RFP plasmids were acquired from Origene (Rockville, MD, USA) [32]. The D446E variant was amplified in the *KCNQ1* plasmid using the QuickChange II XL Site-Directed Mutagenesis Kit (Agilent Technologies; Santa Clara, CA, USA) following the manufacturer’s instructions. Primers were designed using the QuikChange Primer Design tool: D446E-FW: 5′-ttctgggggctcgcacgtgatatgggg-3′; D446E-RV: 5′-ccccatatcacgtgcgagcccccagaa-3′. The variant was confirmed by Sanger sequencing.

### 4.4. Cell Culture and Transfection

Human embryonic kidney 293 cells (HEK293) were used as heterologous system expression for functional characterization. All reagents were purchased from Sigma Aldrich (St. Louis, MO, USA) unless otherwise indicated. Cells were maintained in Dulbecco’s modified Eagle’s medium (DMEM) supplemented with 10% FBS (Hyclone Laboratories Inc., Logan, UT, USA), 100 U/mL penicillin and 100 μg/mL streptomycin (Gibco; Waltham, MA, USA), in a humidified 5% CO_2_ atmosphere at 37 °C. Cells at 60–80% of confluence were used to transiently transfect both channel complex subunits (*KCNQ1* and *KCNE1*) in a 1:1 ratio, using 1.5 μg of each construct. Transfection was performed with the Lipofectamine^TM^ LTX Reagent with PLUS^TM^ Reagent (Invitrogen; Carlsbad, CA, USA) according to the manufacturer’s instructions. Twenty-four hours after transfection, cells were seeded on poly-D-lysine-coated glass coverslips at 2 × 10^4^ cells. Electrophysiological recordings were performed 2 h after the cells were seeded to ensure adhesion to the coverslip.

### 4.5. Electrophysiology

#### 4.5.1. Genotypes

Three different combinations of *KCNQ1* and *KCNE1* plasmids were transfected to HEK293 cells to record the WT-homozygous (446 D/D, *n* = 24), D446E-heterozygous (446 D/E, *n* = 12) and 446E-homozygous (446 E/E, *n* = 15) slow potassium currents (IKs). To compare the role of PKA signaling in IKs among the different *KCNQ1* genotypes, 8Br-cAMP (an activator of cyclic AMP-dependent protein kinase) was added to the intracellular solution, recording voltage-gated currents from eight 446 D/D homozygous or WT, nine 446 D/E heterozygous and seven IKs 446 E/E homozygous transfected cells. All recordings were performed 24 h after transfection. The IKs recordings presented in this study resulted from more than 2 repeated transfections for each genotype. 

#### 4.5.2. Recordings

Whole-cell recordings were made at room temperature with 2–4 MΩ resistance borosilicate pipettes in our recording solutions. For all experiments, the bath (extracellular) solution was (in mM): 145 NaCl, 5 KCl, 1.3 CaCl_2_, 1 MgCl_2_, 0.7 NaH_2_PO_4_ and 10 HEPES; plus ∼6.5 mM NaOH to bring the pH to 7.4 and the osmolality to ∼295 mmol/Kg. The pipette (intracellular) solution was a based KCl solution with CsCl added to reduce the K-driving force as follows (in mM): 100 KCl, 35 CsCl, 0.1 CaCl_2_, 3.5 MgCl_2_, 3 Na_2_ATP, 5 creatine phosphate (Na salt), 0.1 NacAMP, 0.1 Li-GTP, 5 EGTA and 10 HEPES, plus 28 mM KOH to bring the pH to 7.3 and the osmolality to ∼300 mmol/Kg. 8Br-cAMP (200 µM) was added to the intracellular solution to activate PKA-signaling and mimic β-receptor activation. The patch clamp amplifier Multiclamp-700B and DA/A-D converter Digidata 1550A (Molecular Devices; San Jose, CA, USA) were controlled by pClamp 10.5 (Molecular Devices). Capacitive currents were electronically nulled. Series resistances ranged from 2 to 21 MΩ (mean 8.16 ± 0.45 MΩ, *n* = 73) and were 80% compensated for a mean residual value of ∼1.68 MΩ. Potentials were corrected for a liquid junction potential of 5.5 mV, calculated with the JPCalc tool [33] as implemented by Clampex 10.5 (Molecular Devices). Cells were held at −85.5 mV. Nontransfected cells were routinely recorded to assess the possible effect of overlapping Kv currents from endogenous HEK-293 cell ion channels. No inward or outward currents were recorded from these nontransfected cells.

#### 4.5.3. Analysis

IKs voltage dependence was quantified by building activation (conductance–voltage) curves from currents recorded in voltage-clamp mode. Because 8Br-cAMP shifts IKs voltage dependence to more negative potentials, 2 protocols were used to obtain these activation curves. From a holding potential of −85.5 mV, the protocol comprised iterated series of 5 s steps from −45.5 to 74.5 mV in the absence of 8Br-cAMP, or 5 s steps from −65.5 to 34.5 mV in the presence of 8Br-cAMP; a 1 s step to −45.5 mV to measure tail currents; and a 400 ms step to return to the holding potential.

For the IKs voltage dependence analysis, steady-state conductance values were calculated from tail currents when voltage was stepped to −45.5 mV, divided by the driving force (difference between commanded voltage and the K^+^ reversal potential), averaged across all cells, plotted against the test step voltage, and fitted with the Boltzmann function (Equation (1)) as follows:(1)GV=Gmin−Gmax1+eV−V1/2S+Gmax
where *G*(*V*) is the conductance at voltage *V*, *G_min_* and *G_max_* are the minimum and maximum conductance, *V*_1/2_ is the voltage corresponding to the half-maximal activation, and *S* is the voltage corresponding to an e-fold increase in *G*(*V*) (voltage sensitivity). Curve-fitting and statistical analyses were performed with OriginPro 2020b software (OriginLab; Northampton, MA, USA). The parameters of the curve fits (*V*_1/2_, *G_max_*, *S*) were compared for all experimental series. *G_max_* represents the maximum conductance density.

For deactivation kinetics, a tail current at −125.5 mV (preceded by a 5 s voltage step to 34.5 mV) was fitted by a monoexponential function (Equation (2)) to calculate the deactivation time constant (*τ*_deactivation_) in ms. The deactivation describes the return of the IKs channel to its closed state from the open conformation.
(2)ft=∑i=1nAie−tτi+C

The fit solved for the amplitude *A*, the time constant *τ*, and the constant y-offset *C* for each component *i*.

Activation rise time calculations were made with OriginPro 2020b software from 10 to 90% of each step height of the iterated voltage steps of the activation protocol using the linear search algorithm [32].

### 4.6. Statistical Analysis

Data are expressed as mean ± SE. According to the Shapiro–Wilk test, all data sets followed a normal distribution (*p* > 0.19). The significance of differences between means was assessed by a two-way-ANOVA followed by post hoc Tukey’s tests of significance. Differences in allele frequencies were compared using the two-tailed square χ-test.

### 4.7. Protein Modeling of KCNQ1 (Kv7.1)-D446E

In our previous publication we explored *KCNQ1* mutations using homology modeling of the channel [11]. The file ID P51787 from UniprotKB released in 2023 [34] identical to ENST155840 in Ensembl [35] was used as the Kv7.1 channel sequence. Structures from PDB7XNK and 7XNN were used as scaffolds for attached and detached channel conformations. These structures, recently determined by high-resolution cryoelectron microscopy, comprise the full-length human KCNQ1–calmodulin complex [13]. Domain 5 (D5, from 397 to 500 residues) was constructed by threading, short-strand homology and ab initio modeling by Phyre2 [36] because there are no homologous structures available. All other modeling, structural edition and visualization tasks were performed with Molecular Operating Environment (MOE, www.chemcomp.com accessed on 25 August 2023) and a CHARMM27 force field with a distance-dependent description of electrostatics [37]. The builder option of this package was used to model nitrosylation sites and variant structures. Minimizations were carried out until the average energy gradient reached 0.05 kcal/(mol·Å). All structures were validated, and no stereochemical drawbacks were observed. Conformer comparisons between tetramers were performed by matching the alpha carbons of membrane helices.

## Figures and Tables

**Figure 1 ijms-25-00953-f001:**
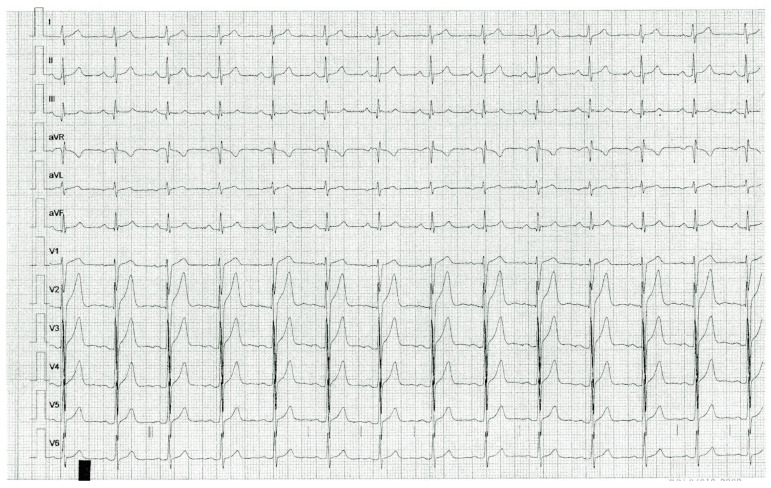
Resting ECG of index case 1, *KCNQ1* p.D446E heterozygous. Early repolarization manifested as terminal QRS slurring and J point elevation in V2, V3, V4 and V5.

**Figure 2 ijms-25-00953-f002:**
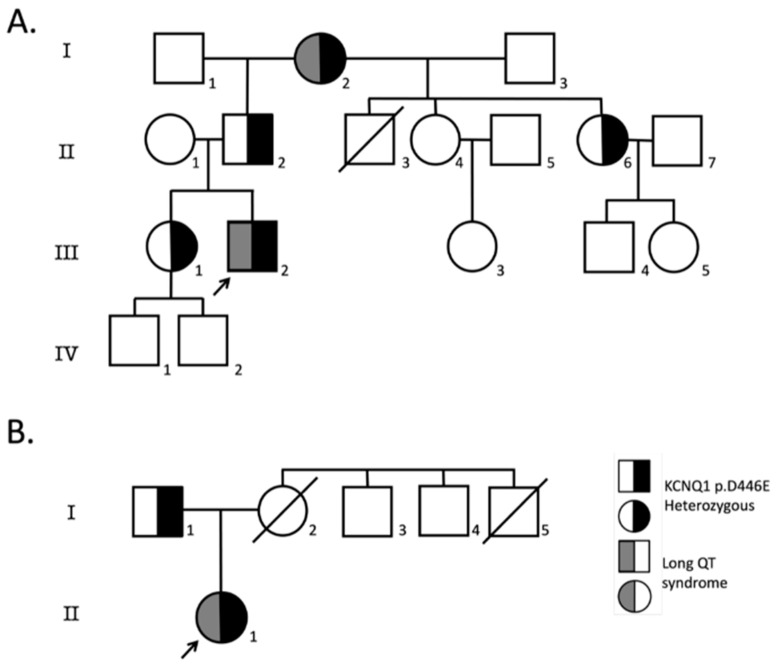
Family pedigrees. Pedigrees of probands heterozygous for *KCNQ1*-D446E variant where at least one first-degree relative was available for screening; (**A**) index case 1, (**B**) index case 2. Index cases are indicated with an arrow, circles indicate females, squares males and symbols with diagonal lines deceased individuals. Symbols half-filled in black are KCNQ1-D446E heterozygous, and symbols half-filled with dark gray have LQTS.

**Figure 3 ijms-25-00953-f003:**
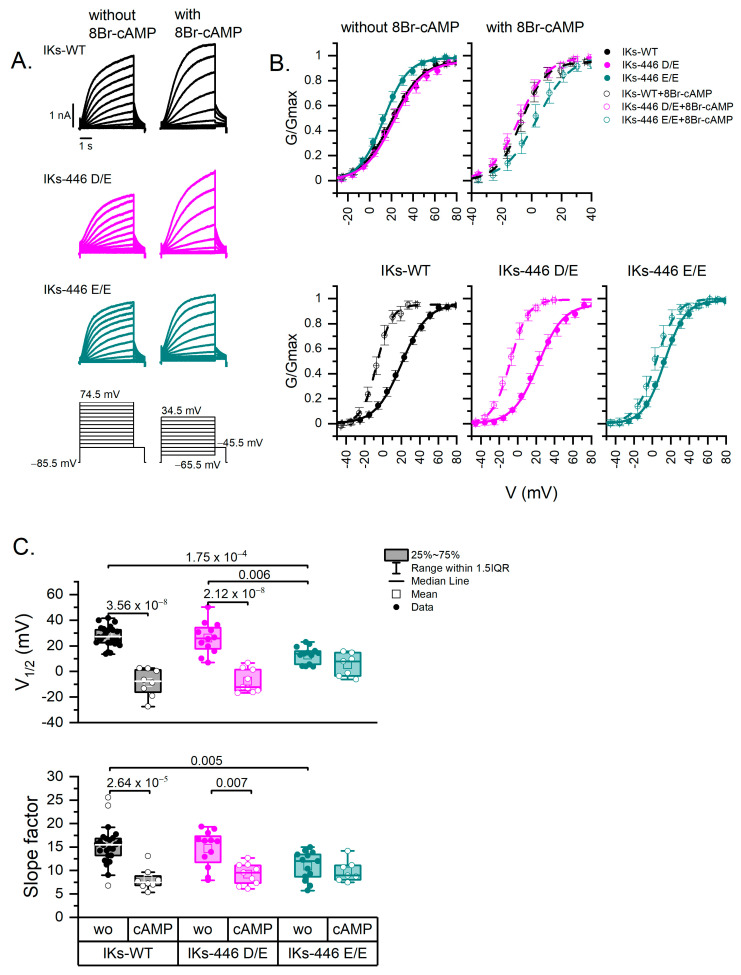
Activation curves. (**A**) Representative IKs currents of all genotypes without and with 8Br-cAMP and voltage-clamp protocols. Holding potential was −85.5 mV for all cells; however, different voltage step ranges are indicated, as activation curves were obtained at different voltage ranges in the presence or absence of 8Br-cAMP. (**B**) Activation curves (G-V) illustrating the variance in voltage dependence (V_1/2_, midpoint of Boltzmann fit) and voltage sensitivity (slope factor) of experimental series without (line-fit) and with (dash line-fit) 8Br-cAMP. Data points are mean values ± SE. (**C**) Box plots showing the differences and distribution of Boltzmann fit parameters; each point represents one cell. Data without and with 8Br-cAMP are represented as filled and empty circles, respectively. Differences were assessed using a two-way ANOVA followed by a post hoc Tukey test of significance. Actual *p* values for comparisons are indicated above the box plots. The number of cells was as follows: WT, *n* = 24; 446 D/E, *n* = 12; 446 E/E, *n* = 15; 446 D/D-cAMP, *n* = 8; 446 D/E-cAMP, *n* = 9; 446 E/E-cAMP, *n* = 7. wo: without 8Br-cAMP; cAMP: with 8Br-cAMP.

**Figure 4 ijms-25-00953-f004:**
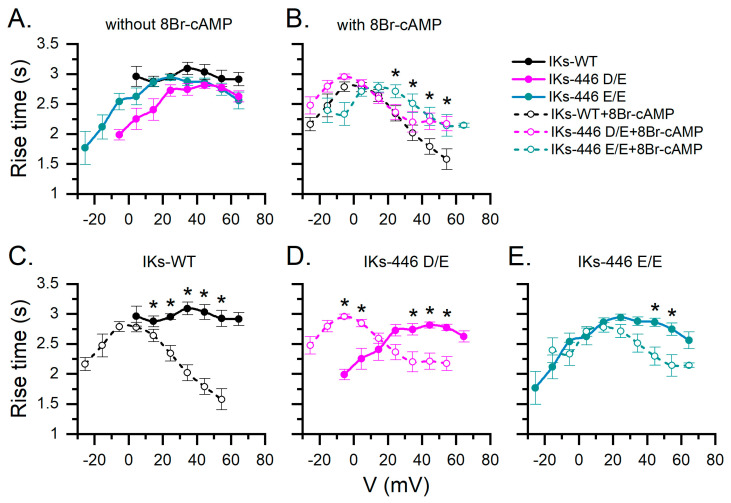
Mean rise time estimations of all IKs series according to voltage without and with 8Br-cAMP stimulation in different genotypes. (**A**) Comparison among genotypes without 8Br-cAMP. (**B**) With 8Br-cAMP. (**C**–**E**) Effect of 8Br-cAMP on rise time for all tested potentials in each genotype. Data without and with 8Br-cAMP are represented as filled and empty circles, respectively. Asterisks indicate significant differences without and with 8Br-cAMP as assessed by a two-way ANOVA followed by a post hoc Tukey test of significance. Number of cells was as follows: WT, *n* = 24; 446 D/E, *n* = 12; 446 E/E, *n* = 15; 446 D/D-cAMP, *n* = 8; 446 D/E-cAMP, *n* = 9; 446 E/E-cAMP, *n* = 7.

**Figure 5 ijms-25-00953-f005:**
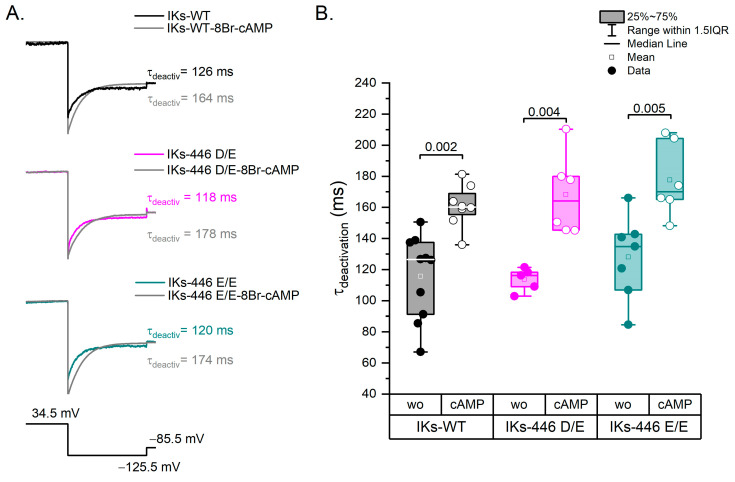
Deactivation kinetics. (**A**) Representative traces of each experimental series showing the IKs deactivation rate from 34.5 mV to −125.5 mV. The prepulse to activate IKs was 5 s at 34.5 mV and the test pulse to deactivate was 1 s at −125.5 mV. (**B**) Box plots showing statistically significant differences between experimental series without and with 8Br-cAMP. Filled circles represent data without 8Br-cAMP and empty circles with the 8Br-cAMP effect. Wo: without 8Br-cAMP; cAMP: with 8Br-cAMP. Differences were assessed using a two-way ANOVA followed by a post hoc Tukey test of significance. WT *n* = 10; WT-cAMP *n* = 8; Homo *n* = 7; Homo-cAMP *n* = 6; Hetero *n* = 5; Hetero-cAMP *n* = 6.

**Figure 6 ijms-25-00953-f006:**
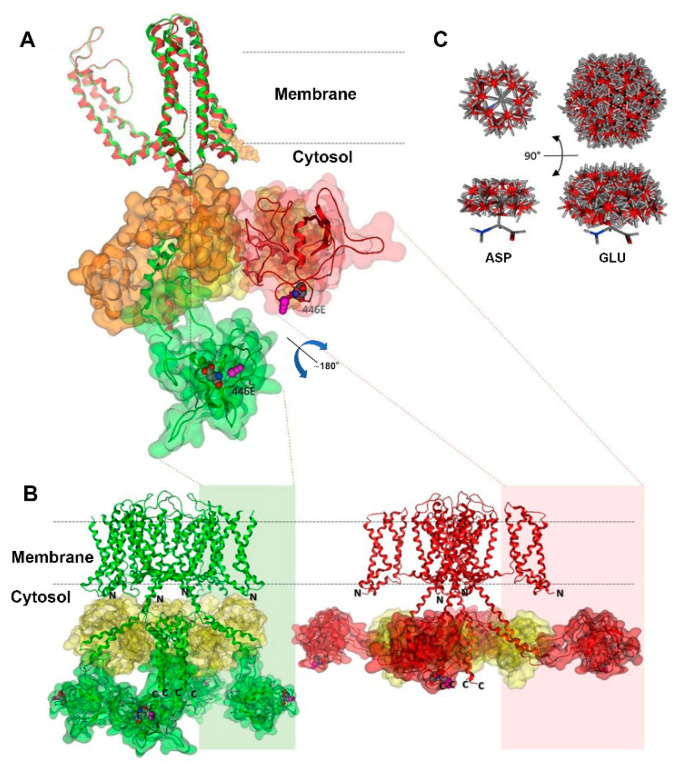
Structural model of the Kv7.1 channel. The vertical dotted line shows the channel axis. The attached conformation is represented in green, while the detached conformation is shown in red. The amino and carboxyl domains, whose experimental structures have not been determined, are denoted by N and C, respectively. (**A**) Superposition of Kv7.1 monomers highlighting the dynamic movement of our modelled domain 5 upon detachment of the VSD domain. Domain 5 is delineated by green or red outlines, while calmodulin is depicted in orange or yellow. Notably, domain 5 changes both its position and orientation, as evident by the positioning of the variant site and its adjacent nitrosylation site (magenta) in the different conformations. (**B**) Kv7.1 tetramers in attached (**left**) and detached (**right**) conformations, with the same color codes as in panel A except for calmodulin, which is yellow in both cases. The sizes of the amino and carboxyl domains are comparable to that of domain 5. N and C represent the locations of the chains of each domain within the cytosol. (**C**) Comparison of upper and lateral views showcasing the conformational differences between aspartic and glutamic acid side chains. These differences are generated by the rotation of single bonds at 60° intervals, while maintaining fixed backbone atoms, illustrating variations in the volume of their potential conformations.

## Data Availability

The data presented in this study can be found in online repositories at https://www.ncbi.nlm.nih.gov/snp/ (accessed on 7 November 2023), rs199472780.

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
