# Peer review of "(untitled)"

_ijms, 2024, doi:10.3390/ijms25020953_

Round 1
Reviewer 1 Report
Comments and Suggestions for Authors
In this study, the investigators provide the experimentally observed and in silico results showing the influence of the KCNQ1 p.D446E variant on cardiac IKs (KCNQ1) expressed in HEK293 cells. Regarding this work, several comments, including experimental procedures, are shown as follows.
(1) In Abstract (line 64, the sentence should be changed to “… either gain or loss of function…”. Similarly, the title should be appropriately changed to “…gain or loss…”
(2) KCNQ1-encoded current is characterized by slow activation and deactivation properties. In lines 467-472, why did the investigators not measure the activation kinetics, such as activation time constant of IKs. In addition to an increase in IKs density, the elevation of cytosolic cyclc AMP may decrease the activation time constant of IKs. The description of ‘voltage sensitivity’ shown in line 462 may not be appropriate.
(3) From a broad perspective, while the slope factor of the quasi-activation curve (conductance versus membrane voltage relationship of the current) remains unchanged, a change in the V1/2 value can also alter voltage sensitivity. Strictly speaking, the slope factor pertains to sensitivity to changes in apparent gating charge (i.e., q value) of the curve. Perhaps according to the Boltzmann distribution, it is necessary to calculate q values in each activation curve in the absence and presence of 200 mM 8Br-cAMP perfusion, e.g., the results in Figure 1B.
(4) In lines 434, in this study, 8Br-cAMP at a concentration of 200 mM was included in the recording pipette. The experimentally observed results in Figures 1A and 3A between the procedures conducted without and with 8Br-cAMP did not appear to display the results in the same cell. The comparison between them should therefore be rather difficult. Please comment on this discrepancy in different cells examined. Please include this issue into the Discussion section of the manuscript.
(5) If a comparison is to be made between different cells, using current density for comparison would be more reasonable. This involves excluding the interference of whole-cell capacitance.
(6) In the entire text of the manuscript, it is necessary to explicitly explain the experimental procedures for ‘without or with 8Br-cAMP’. Additionally, please clarify how the comparisons were conducted and how the p-values were calculated in each set of the statistical analysis.
(7) It is better to perform effects of extracellular membrane-permeant cAMP analog on IKs for comparison in the same cell. Different cells in the size and expression are not easy for comparisons conducted between with and without the inclusion of 8BrcAMP in the intracellular solution.
(8) In Figures 1C and 3C, please include the number of cells examined during each set of experiments, along with the detailed statistical methods used (such as ANOVA and t-test).
(9) In Figure 3A, the deactivating tail current traces were almost overlapped and superimposed. Please make an expanded record to show clear difference in the deactivation time course of IKs, although the value of tinactiv was shown.
(10) In Figure 3B, despite the visually appealing graphics, please provide the N values, along with the statistical methods (e.g., ANOVA or t-tests) and p-values, as different cells were being compared. Please include N values in each set of comparisons appearing in text of the manuscript.
(11) In Figure 3A, please show the duration applied during the voltage-clamp protocol.
(12) In lines 356-358, the statement about ‘…larger volume and increased solvent-exposed negative charge….’ Is quite confusing to some extent. Please thus revise it and elaborate it more clearly in a scientific way.
(13) If possible, could the blocker of IKs need to be tested in order to verify the existence of KCNQ1-encoded currents with no overlapping of Kv11 currents? Please note that many endogenous ion channels are expressed in HEK-293 cells (Zhang et al., Pflugers Arch 2022;474(7):665-680).
(14) Please explain how to assess the gain or loss of function in IKs in this study. The statement also needs to be included in the revised manuscript. Is the function of IKs based on the basal density of IKs or the responsiveness of expressed IKs density to 8Br-cAMP?
(15) The data in the Supplementary material could be interesting and need to be included in the main portion of the manuscript.
Comments on the Quality of English Languagenon-available
Author Response
Response to Reviewer 1
Our sincere thanks for taking the time to review our manuscript and for your comments that have helped improve our manuscript.
- In Abstract (line 64, the sentence should be changed to “… either gain or loss of function…”. Similarly, the title should be appropriately changed to “…gain or loss…”
The function of voltage-gated channels is indeed complex. Most studies characterizing the functional consequences of mutations analyze current and gating properties, and very few analyze the response to cAMP or other stimuli. For the Nav1.5 sodium channel, there are examples of mutations with complex electrophysiological phenotypes that simultaneously cause a hyperpolarizing shift (GOF) and decrease the peak current (LOF), while others have been reported to decrease the peak of the early current (LOF) and increased late current (GOF) (Akai, J. et al., FEBS Lett. 2000, 479, 29–34; Kapplinger et al., Circ. Cardiovasc. Genet. 2015, 8, 582–595.; Medeiros-Domingo et al. Heart Rhythm 2009, 6, 1170–1175.; Nakajima et al., Heart Rhythm 2015, 12, 2296–2304. doi.org/10.1016/j.hrthm.2015.05.032).
Similarly, a combined GOF/LOF consequence has been described for the KCNQ1 A300T mutation, with GOF caused by a hyperpolarizing shift and LOF by moderately decreasing Kv7.1 conductance (Antúnez-Argüelles et al. (2017) 5;627:40-48; Priori et al., Circulation. (1998), 97:2420–2425; Bianchi et al., (2000), Am. J. Physiol. Heart Circ. Physiol. 279, H3003–H3011; Kang et al (2008), Biochemistry (Mosc) 47, 7999–8006; Smith et al. (2007), Biochemistry (Mosc) 46,14141–14152).
The electrophysiological characterization of 446E revealed both a hyperpolarizing shift (GOF) and a lack of response to c-AMP (LOF), thus the preposition “or” would be not quite accurate. For the sake of clarity, we changed the title of the manuscript to describe the consequences as a complex electrophysiological phenotype and have clarified this complex mixed phenotype in the text. The title now reads: “KCNQ1 p.D446E variant as a risk allele for arrhythmogenic phenotypes: electrophysiological characterization reveals a complex phenotype affecting IKs voltage dependence by causing a hyperpolarizing shift and a lack of response to protein kinase A activation”. Moreover, we have added the following statement to the discussion section: Thus, while in basal conditions the hyperpolarizing shift can be interpreted as gain of function, under 8Br-cAMP stimulation there is a loss of function.
- KCNQ1-encoded current is characterized by slow activation and deactivation properties. In lines 467-472, why did the investigators not measure the activation kinetics, such as activation time constant of IKs. In addition to an increase in IKsdensity, the elevation of cytosolic cyclc AMP may decrease the activation time constant of IKs. The description of ‘voltage sensitivity’ shown in line 462 may not be appropriate.
We agree that measurements of activation kinetics are crucial to assess cAMP effects on IKs. We measured activation kinetics as rise time, plotted in Figure 2. We decided to assess activation kinetics by measuring rise time because the classical exponential fit (we tested one and multiple terms) did not describe the shape of IKs. Other authors have used rise time to assess kinetics: Aggarwal et al., 2023; Nat Methods 20, 925–934 (2023). https://doi.org/10.1038/s41592-023-01863-6);.Clements et al., 1998; The Journal of Neuroscience, 18(1):119–127); McClellan and Twyman, 1999; Journal of Physiology, 515.3, pp. 711727).
In addition, we observed that cAMP, contained in the recording pipette, decreased activation time of IKs in WT phenotype (Figure 2C) but had complex effects in hetero and homozygous phenotypes (Figure 2D and 2E).
- From a broad perspective, while the slope factor of the quasi-activation curve (conductance versus membrane voltage relationship of the current) remains unchanged, a change in the V1/2value can also alter voltage sensitivity. Strictly speaking, the slope factor pertains to sensitivity to changes in apparent gating charge (i.e., q value) of the curve. Perhaps according to the Boltzmann distribution, it is necessary to calculate q values in each activation curve in the absence and presence of 200 mM 8Br-cAMP perfusion, e.g., the results in Figure 1B.
We agree that a change in the V1/2 value can also alter voltage sensitivity. In fact, we observed a change in V1/2 and voltage sensitivity in WT and D446E heterozygous cells in response to 8Br-cAMP as slope factor values decreased when 8Br-cAMP was contained in the recording pipette. However, we did not observe a change in V1/2 and voltage sensitivity in 446E homozygous cells, suggesting that 446E homozygosity impedes the response to c-AMP (loss of function in response to adrenergic stimulation). We are unsure if we have correctly understood the concern that is being raised in numeral (3).
- In lines 434, in this study, 8Br-cAMP at a concentration of 200 mM was included in the recording pipette. The experimentally observed results in Figures 1A and 3A between the procedures conducted without and with 8Br-cAMP did not appear to display the results in the same cell. The comparison between them should therefore be rather difficult. Please comment on this discrepancy in different cells examined. Please include this issue into the Discussion section of the manuscript.
We agree that ideally comparisons should have been made in the same cell. We have commented on this discrepancy as a limitation of the study in the discussion section.
- If a comparison is to be made between different cells, using current density for comparison would be more reasonable. This involves excluding the interference of whole-cell capacitance.
We did in fact consider cell capacitance, as stated in line 157: “No differences in terms of conductance density (G/Cm) were observed among genotypes (WT or 446 D/D, 446 D/E and 446 E/E) (p > 0.56)”; and line 165: “No mean differences in conductance density in response to 8Br-cAMP were observed among genotypes (p > 0.43).” We decided to show G/Gmax plots and not G/Cm for the sake of clarity.
- In the entire text of the manuscript, it is necessary to explicitly explain the experimental procedures for ‘without or with 8Br-cAMP’. Additionally, please clarify how the comparisons were conducted and how the p-values were calculated in each set of the statistical analysis.
All differences in this manuscript were assess by a Two-way ANOVA followed by post-hoc Tukey test of significance, we used Origin software to calculate these values. Actual p-values for all statistical differences indicated in the figures (box plots). We have included number of cells in the figure legends, as suggested.
- It is better to perform effects of extracellular membrane-permeant cAMP analog on IKsfor comparison in the same cell. Different cells in the size and expression are not easy for comparisons conducted between with and without the inclusion of 8BrcAMP in the intracellular solution.
We agree comparisons in the same cell is best. However, we used intracellular and not extracellular cAMP for testing, in order to have better control of the c-AMP experimental concentrations and how they affected IKs. With this experimental approach, we tested statistical differences accordingly with a two-way-ANOVA with a post-hoc Tukey’s test of significance. Tested factors were genotype and absence of presence of 8Br-cAMP. Testing 8Br-cAMP in different cells and not in the same cell has been acknowledged as study limitation in the discussion section.
- In Figures 1C and 3C, please include the number of cells examined during each set of experiments, along with the detailed statistical methods used (such as ANOVA and t-test).
All differences in this manuscript were assess by a Two-way ANOVA followed by post-hoc Tukey test of significance. Actual p-values are above all statistical differences indicated in box plots. We have included number of cells in the figure legend. Because 2 figures from supplementary material were included in the main manuscript, Figure 1C is 3C, and Figure 3C is 5C in the revised version.
- In Figure 3A, the deactivating tail current traces were almost overlapped and superimposed. Please make an expanded record to show clear difference in the deactivation time course of IKs, although the value of tinactivwas shown.
We have expanded the tail currents to clarify differences in deactivation time course and have included the test pulse duration in the legend (5 s). Figure 3 A is Figure 5A in the revised manuscript.
- In Figure 3B, despite the visually appealing graphics, please provide the N values, along with the statistical methods (e.g., ANOVA or t-tests) and p-values, as different cells were being compared. Please include N values in each set of comparisons appearing in text of the manuscript.
All differences in this manuscript were assess by a Two-way ANOVA followed by post-hoc Tukey test of significance. Actual p-values are above all statistical differences indicated in box plots. We have included statistical description and the number of cells in the figure legends.
- In Figure 3A, please show the duration applied during the voltage-clamp protocol.
Duration step (5 s) is mentioned in methods. We have added duration step and tail in the figure legend. Figure 3A corresponds to Figure 5A in the revised manuscript.
- In lines 356-358, the statement about ‘…larger volume and increased solvent-exposed negative charge….’ Is quite confusing to some extent. Please thus revise it and elaborate it more clearly in a scientific way.
We revised this statement, which no reads: “Summarizing, our model predicts that the interactions of the 466E negative charge with K+ ions and other groups in the surrounding environment may be stronger than those of the WT aspartate” hoping it is now clear.
- If possible, could the blocker of IKsneed to be tested in order to verify the existence of KCNQ1-encoded currents with no overlapping of Kv11 currents? Please note that many endogenous ion channels are expressed in HEK-293 cells (Zhang et al., Pflugers Arch 2022;474(7):665-680).
We confirmed that expression of endogenous ion channels in HEK-293 cells under our experimental conditions is negligible. We routinely recorded from “empty” or non-transfected HEK cells and observed no inward or outward currents. We have clarified this in the methods section.
- Please explain how to assess the gain or loss of function in IKsin this study. The statement also needs to be included in the revised manuscript. Is the function of IKsbased on the basal density of IKs or the responsiveness of expressed IKs density to 8Br-cAMP?
We have added the following statement to the discussion section: “Thus, while in basal conditions the hyperpolarizing shift can be interpreted as gain of function, under 8Br-cAMP stimulation there is a loss of function.”, hoping to have clarified this matter.
- The data in the Supplementary material could be interesting and need to be included in the main portion of the manuscript.
We have included the Figures S1 and S2 in the manuscript, corresponding to Figures 1 and 2.

Reviewer 2 Report
Comments and Suggestions for Authors
This paper provides a better understanding of a low frequency mutation KCNQ1p.D446E. Based on work presented in the manuscript the mutation is potentially causitive of, or contributive to, long QT syndrome, but currently regarded as a variant of unknown importance. The variant occurs at higher frequencies in Latin populations and the authors suggest that the prevalence may be higher that that reported in public data sets. The authors present a thorough investigation that includes sequencing of patient DNA, expression and biophysical characterization in mammalian cells, and protein modelling. KCNQ1p.D446E was co-expressed with minK in HEK cells. Biophysical properties were studied under several conditions including those reflecting physiological situations for heterozygotic and homozygotic expression (transfection with WT, WT+mutant, mutant alone). The presented work is both significant and substantial although a few issues are noted.
Minor editing suggestions
Pg 4, ln 151. Change "exemplary" to "example"
Conceptual Issues
In cases where allele frequency is low, many sequences are required for comparative data. Given that 2/63 samples identified D466E as a variant, it is interesting but not revelatory that the allele frequency was about 30-fold higher than the referenced dataset
Index case 1 history doesn't make sense as it is written. It includes a 16-year old asymptomatic male who experienced sudden cardiac death at school, while also stating the he was sent to the cardiological institute. Furthermore, details of cardiac examination are reported in this manuscript.
Figures and Legends
Figure 1 - Figure is too small. It is not possible to read the text or to discriminate between lines on graphs.
The bottom of Figure 1A presents examples of two different voltage-clamp protocols. It is unclear why these are different. The organization of the figure implies that different protocols were used in experiments with and without 8Br-cAMP. If this is the case then the results should not be compared as holding Vm's appear to be different (-85.5 mV vs -65.5 mv)
Materials and Methods
Page10 Section 4.3. Clarify how the mutation was established. It sounds like site-directed mutagenesis was used to create the mutation in the KCNQ1 expression plasmid from Origene. If this is the case then remove "cloned" (line 395) and reword. Cloned implies that the entire mutant gene was inserted in the vector.
Author Response
Response to Reviewer 2
Thank you for taking the time to review our manuscript and help improve it.
Minor editing suggestions
Pg 4, ln 151. Change "exemplary" to "example"
Exemplary has been changed to example.
Conceptual Issues
In cases where allele frequency is low, many sequences are required for comparative data. Given that 2/63 samples identified D466E as a variant, it is interesting but not revelatory that the allele frequency was about 30-fold higher than the referenced dataset.
We agree this is not revelatory per se, but the observation of this difference in allele frequencies gave us a reason or justification to perform the functional studies. In fact, we first screened 644 Mexican patients from a population-based study for the D446E variant using a Taqman probe but did not find a single individual with this variant. Because public databases contain tens of thousands of samples from Latinos, we thought it was best to compare the allele frequency with Latinos from Exac database. That is why we stated “We thus analyzed the biophysical phenotype of this variant in the heterozygous and homozygous states, under basal conditions and after protein kinase A (PKA) activation simulating adrenergic stimulation and used protein modelling to understand the possible contribution of this variant to the etiology of arrhythmogenic phenotypes, as what Giudicessi et al. suggest is a functional risk allele”. We hope it is clear that the difference in minor allele frequencies is an observation that led us to test whether the variant has functional consequences.
Index case 1 history doesn't make sense as it is written. It includes a 16-year old asymptomatic male who experienced sudden cardiac death at school, while also stating the he was sent to the cardiological institute. Furthermore, details of cardiac examination are reported in this manuscript.
Indeed, it does not make sense. He suffered aborted sudden death, as he responded to CPR and was referred to the National Institute of Cardiology. This has been corrected in the text which now reads: “Index case 1 was a previously asymptomatic 16-year-old male who suffered generalized seizures followed by cardiac arrest while sitting in a classroom which reverted after resuscitation maneuvers, and was referred to the Instituto Nacional de Cardiología “Ignacio Chávez”.
Figure 1 - Figure is too small. It is not possible to read the text or to discriminate between lines on graphs.
The bottom of Figure 1A presents examples of two different voltage-clamp protocols. It is unclear why these are different. The organization of the figure implies that different protocols were used in experiments with and without 8Br-cAMP. If this is the case then the results should not be compared as holding Vm's appear to be different (-85.5 mV vs -65.5 mv)
We have amplified Figure 1 (Figure 3 in the revised version). All cells were held at the same membrane potential (-85.5 mV), holding potentials did not differ in experiments with and without 8-Br-cAMO. The figure shows different voltage step ranges with and without 8Br-cAMP for clarity, as activation curves were obtained at different ranges in the presence and absence of 8Br-cAMP. This was clarified in the figure legend. Because another reviewer requested supplementary figures be included in the main text, Figure 1 corresponds to Figure 3 in the revised manuscript.
Materials and Methods
Page10 Section 4.3. Clarify how the mutation was established. It sounds like site-directed mutagenesis was used to create the mutation in the KCNQ1 expression plasmid from Origene. If this is the case then remove "cloned" (line 395) and reword. Cloned implies that the entire mutant gene was inserted in the vector.
Indeed, this is a mistake, we did not clone the variant. We have corrected the text to state that the vector with the variant was amplified.

Reviewer 3 Report
Comments and Suggestions for Authors
The study titled "KCNQ1 p.D446E variant as a risk allele for arrhythmogenic phenotypes: electrophysiological characterization reveals both gain and loss of Kv7.1 function" by González-Garrido et al is very well-designed study assessing the function of variant mutation in KCNQ1 gene and its possible risk in cardiac arrhythmogenicity. They have characterized the WT and heterozygous (D/E) and homozygous (E/E) mutants in HEK293 cells (heterologous expression system) for their electrophysiology and with adrenogenic stimulation. They also used computational modelling to analyze the structural changes that could impact the normal function of the channel. The results are presented well and the conclusions are acceptable. These are just initial characterizations, and they obviously require more meticulous studies in systems that are closer to cardiac cell and in vivo validations in the long term. However, this study can be considered a first step towards the better understanding of these mutant variants in KCNQ1 channels.
I felt that even though the authors explicitly state the statistics in their method section, it would be better to include the statements in each of the figure legends with the number of cells analyzed per group and the statistical test performed to get the p values. The study is limited only to one cell type which is a general cell type (HEK293) therefore the study results may not exactly reflect a cardiac cell type. Additionally, the title may be modified to this end clearly stating their finding and not being biased. The title should be modified to include exact findings with “…gain or loss…” .
Author Response
Response to Reviewer 3
Our sincere thanks for taking the time to review our manuscript and help improve it.
I felt that even though the authors explicitly state the statistics in their method section, it would be better to include the statements in each of the figure legends with the number of cells analyzed per group and the statistical test performed to get the p values.
We have included this description and the number of cells analyzed in the figure legends, as suggested. Because another reviewer requested supplementary figures be included in the main text, Figure numbers have changed in the revised manuscript.
The study is limited only to one cell type which is a general cell type (HEK293) therefore the study results may not exactly reflect a cardiac cell type.
We agree and have acknowledged this as a limitation of the study in the discussion section.
Additionally, the title may be modified to this end clearly stating their finding and not being biased. The title should be modified to include exact findings with “…gain or loss…” .
The electrophysiological characterization of 446E revealed both a hyperpolarizing shift (GOF) and a lack of response to c-AMP (LOF), thus the preposition “or” would be not quite accurate. According to your suggestion, we changed the title of the manuscript to describe the consequences as a complex electrophysiological phenotype and have clarified this complex mixed phenotype in the text. The title now reads: “KCNQ1 p.D446E variant as a risk allele for arrhythmogenic phenotypes: electrophysiological characterization reveals a complex phenotype affecting IKs voltage dependence by causing a hyperpolarizing shift and a lack of response to protein kinase A activation”. Moreover, we have added the following statement to the discussion section: “Thus, while in basal conditions the hyperpolarizing shift can be interpreted as gain of function, under 8Br-cAMP stimulation there is a loss of function.”

Round 2
Reviewer 1 Report
Comments and Suggestions for Authors
The authors have addressed the important questions raised by the reivewer. It is an interesting paper.